# Feasibility of Utilizing Social Media to Promote HPV Self-Collected Sampling among Medically Underserved Women in a Rural Southern City in the United States (U.S.)

**DOI:** 10.3390/ijerph182010820

**Published:** 2021-10-14

**Authors:** Matthew Asare, Beth A. Lanning, Sher Isada, Tiffany Rose, Hadii M. Mamudu

**Affiliations:** 1Department of Public Health, Robbins College of Health and Human Sciences Baylor University, One Bear Place, Waco, TX 76798, USA; Beth_Lanning@baylor.edu (B.A.L.); Sher_isada1@baylor.edu (S.I.); Tiffany_mason@baylor.edu (T.R.); 2Department of Health Services Management and Policy, College of Public Health, East Tennessee State University, Johnson City, TN 37614, USA; MAMUDU@mail.etsu.edu

**Keywords:** social media, HPV self-testing, medically underserved women, facilitators and barriers to social media use

## Abstract

Background: Social media (Facebook, WhatsApp, Instagram, Twitter) as communication channels have great potential to deliver Human papillomavirus self-test (HPVST) intervention to medically underserved women (MUW) such as women of low income. However, little is known about MUW’s willingness to participate in HPVST intervention delivered through social media. We evaluated factors that contribute to MUW’s intention to participate in the social media-related intervention for HPVST. Methods: A 21-item survey was administered among women receiving food from a local food pantry in a U.S. southern state. Independent variables were social media usage facilitators (including confidentiality, social support, cost, and convenience), and barriers (including misinformation, time-consuming, inefficient, and privacy concerns). Dependent variables included the likelihood of participating in social-driven intervention for HPVST. Both variables were measured on a 5-point scale. We used multinomial logistic regression to analyze the data. Results: A total of 254 women (mean age 48.9 ± 10.7 years) comprising Whites (40%), Hispanics (29%), Blacks (27%), and Other (4%) participated in the study. We found that over 44% of the women were overdue for their pap smears for the past three years, 12% had never had a pap smear, and 34% were not sure if they had had a pap smear. Over 82% reported frequent social media (e.g., Facebook) usage, and 52% reported willingness to participate in social media-driven intervention for HPVST. Women who reported that social media provide privacy (Adjusted Odds Ratio (AOR) = 6.23, 95% CI: 3.56, 10.92), provide social support (AOR = 7.18, 95% CI: 4.03, 12.80), are less costly (AOR = 6.71, 95% CI: 3.80, 11.85), and are convenient (AOR = 6.17, 95% CI: 3.49, 10.92) had significantly increased odds of participating in social media intervention for HPVST. Conclusions: The findings underscore that the majority of the MUW are overdue for cervical cancer screening, regularly use social media, and are willing to participate in social media-driven intervention. Social media could be used to promote HPV self-testing among MUW.

## 1. Introduction

Human Papillomavirus (HPV) tests and pap tests have contributed to the decline of the incidence of cervical cancer. The American Academy of Family Physicians and the U.S. Preventive Services Task Force recommend cytology (pap smear) and HPV tests for women within different age groups [1,2]. Women between 21 and 29 years of age are recommended to be screened every 3 years with cytology alone [2,3]. Those between 30 and 65 years of age are recommended to be screened every 5 years with cytology plus HPV testing or every 3 years with cytology alone [1,2]. While a pap test (cytology) requires a physician to obtain samples from the cervix for further examination [4,5], the HPV tests require samples from the cervix but can be obtained using brushes or swabs or other devices by either physician or by self-screening [4,5]. Non-participation is the fundamental reason for the persistent cervical cancer cases in women who qualify for screening under the current guidelines [3,6]. The most vulnerable populations (herein defined as medically underserved women (MUW) such as low-income women (LIM)) are underrepresented in the most widely known physician-performed HPV tests. An estimated 14 million women in the U.S. have not been screened and the majority of them are low-income [7]. Overall, 81% of women are up to date with the screening in the U.S., which is below the Healthy People 2030 stated goal of 84% [8,9].

Barriers to physician-performed cervical cancer screening include cost, embarrassment, the anticipation of pain, male physician presence, lack of knowledge about screening, language barriers, other health issues, transportation, forgetting to schedule appointments, and lack of time [10,11,12,13,14,15,16]. HPV self-screening could help address provider-related barriers to cervical cancer screening. Self-collection of vaginal samples is a method in which women collect samples themselves and send them to the clinic or laboratory for testing. HPV self-collection is convenient, may increase women’s sense of privacy and improve access in remote areas, and may decrease stigma and embarrassment. Several studies have demonstrated that HPV self-tests have a high sensitivity in the diagnosis of cervical (pre)cancer [17,18,19,20]. The use of mobile technology as a communication healthcare channel has the potential to help promote HPV self-screening.

### Mobile Technology

Mobile health(mHealth) technology, including the use of mobile phone apps, social media (Facebook, WhatsApp, Twitter), text-messaging, e-mail, and phone calls, has been efficacious in delivering cancer screening interventions to women [21,22,23]. There is compelling evidence that healthcare professionals and the general public are using mobile technology successfully as a communication tool for healthcare decisions [24,25]. Several studies support the use of a mobile app to increase breast cancer screening [21,22,23]. Additionally, mobile phone usage is high in the U.S., with 96% of adults owning cell phones [26]. The mHealth technology has the potential to reduce the cost of health care and improve health outcomes. These technologies can support continuous health monitoring at both the individual and population level, encourage healthy behaviors that can prevent or reduce health problems, support chronic disease self-management, enhance provider knowledge, reduce the number of healthcare visits, and provide personalized, localized, and on-demand interventions in ways previously unimaginable [27,28,29]. However, the use of mobile technology to promote a cervical cancer screening program among medically underserved women (e.g., LIW) has been understudied, partly due to the assumption that MUW have limited access to mobile technology [26]. Understanding the frequency of MUW’s social media usage and HPV self-testing behaviors and identifying factors influencing their social media usage will help in developing a tailored intervention to promote cervical cancer screening among MUW. The primary purpose of this study was to examine social-media usage behavior among MUW in a southern state of the U.S. The secondary purpose was to determine factors that contribute to MUW’s willingness (intention) to participate in HPV self-cervical cancer screening using social media-related intervention. We assessed the participants’ behavioral intention to participate in HPVST intervention as the outcome variable that is consistent with health behavior research [30,31]. Assessing the behavioral intention to participate in HPVST intervention delivered through social media is critical because behavioral intention is the most proximal antecedent of actual behavior [32,33]. Systematic reviews of the literature concluded that intention significantly influences a person’s actual behavior [32,33].

## 2. Materials and Methods

### 2.1. Study Design and Setting

We recruited women visiting a local food pantry in central Texas to participate in a cross-sectional study to assess the willingness to participate in studies involving social media interventions between 5 October and 30 November 2020. The food pantry, located in a southern state, serves thousands of families each year [34].

### 2.2. Sample Size Determination

We used the G*Power software [35] to calculate the sample size for the study. Based on the following parameters: an effect size (f2) of 0.1, an alpha of 0.05, power of 0.95, and 5 predictors, it was determined that a sample size of 204 was needed. However, we increased the sample by 20% to account for any missing data resulting in a final sample of 245 women.

### 2.3. Recruitment Method

We used face-to-face contacts to recruit women who were receiving fresh and canned produce from the local food pantry for the study. Women who were aged 30 years or older and could read and write in English and/or Spanish were included in the study. We gave a hard copy of the questionnaire to each participant to complete. Participants were required to provide informed consent prior to completing the survey and were given a $10 gift card for completing the survey. The study protocol was approved by the university’s Institutional Review Board (IRB reference #1,649,682 and approval date was 26 July 2020).

### 2.4. Measures

After reviewing existing instruments [36] and literature [37,38], a 23-item survey consisting of independent variables (8 items), dependent variables (2 items), social media usage behavior (4 items), and covariates (9 items) was developed.

#### 2.4.1. Dependent Variable

Dependent variables were intention for HPV self-testing and the likelihood of participating in a social media HPV self-screening study. The items were (a) “I intend to take the HPV self-sampling test if I get self-sampling kits”, measured by a 5-point scale with response option strongly disagree and strongly agree, and (b) “if HPV self-sampling test education is provided on any social media platform (Facebook, Instagram, WhatsApp, Twitter, text messaging), I will be … to participate in that program”, measured by a 5-point scale with response option “less likely” to “most likely”.

#### 2.4.2. Independent Variables

The independent variables included facilitators of and barriers to social media utilization. The facilitator items were “I use social media (Facebook, Twitter, Instagram) because they (a) provide privacy (i.e., open to talk about health status because the people may not know you) (b) provide social support (c) are less costly, and (d) are convenient to use”. The barrier items were “I rarely use social media because they (a) provide misinformation, (b) can be time-consuming and distractive, (c) are inefficient to use, and (d) confidentiality concerns”. The facilitator and barrier items were measured on 5-point scales ranging from “strongly disagree to strongly agree”.

#### 2.4.3. Covariates

The covariates were: respondent age, marital status, race, income, employment, health insurance, employment status, knowledge about HPV and cervical cancer, and knowledge about cervical self-screening.

### 2.5. Statistical Analyses

Descriptive statistics, such as frequencies and means, were used to analyze the demographic and other covariate data. A logistic regression model was used to analyze the associations between the independent variables and the dependent variable data. For logistic regression, the independent and dependent variables were dichotomized, where a score between 1–3 was recoded as 0 (No) and 4–5 was recoded as 1 (Yes). We controlled for the covariates in the logistics regression model. The significant result was set a priori at *p*-value < 0.05. All data were analyzed using the IBM Statistical Package for Social Sciences (SPSS 25).

## 3. Results

### 3.1. Demographics

A sample of 254 women (mean age 48.9 ± 10.7 years) comprising Whites (40%), Hispanics (29%), and Blacks (27%) participated in the study. We found that 67.72% of the women reported they had had a Pap smear prior, 12% had never had a Pap smear, and 20.08% were not sure if they had had a pap smear. Approximately 44% of respondents were overdue for their pap smears for the past 3 years. Eighty percent of women in the study reported an annual income below $20,000, 39% reported being uninsured, and 80% were unemployed. Over 82% reported regular social media (e.g., Facebook, text messaging) usage, and 57.48% of the women reported that they had intention to participate in HPV self-testing (Table 1).

### 3.2. Subgroup Analysis

The odds that women between the ages of 30 and 49 years old intended to participate in the social media-related study were 1.81 times the odds that women who were 50 years and above intended to participate in a social media-related study. The odds that widows frequently used social media were 2.96 times the odds that married women used social media. The odds that divorced women intended to participate in HPV self-testing were 2.39 times the odds that married women intended to participate in HPV self-testing. Compared to women who are employed, the unemployed women were more likely to report regular usage of social media (Table 2).

### 3.3. Social Media Facilitators and Barriers

Factors that encouraged women’s social media utilization included beliefs that social media provide privacy (50.79%), social media provide social support (54.72%), social media are less costly (56.30%) and social media usage is convenient (58.27%). Factors that are barriers to social media usage include misinformation (33.07%), time-consuming and distracting (31.10%), insufficient information (35.04%), and confidentiality concerns (53.15%) (Table 3).

### 3.4. Predictors of Social Media Participation

When participants were asked about their willingness to participate in a social media-driven intervention or study, 52.36% of the women reported that they would most likely participate, 27.95% would not, and 19.69% were not sure. More than half (55.12%) of the respondents said that they would be comfortable participating in social media-related interventions or studies (see Table 3). After controlling for the covariates, we found that women’s likelihood of participating in a social media-driven cervical cancer screening study was associated with their perception that social media provide privacy vs. no privacy (Adjusted Odds Ratio (AOR) = 6.23, 95% CI: 3.56, 10.92), social media provide social support vs. no social support (AOR = 7.18, 95% CI: 4.03, 12.80), social media are less costly vs. costly (AOR = 6.71, 95% CI: 3.80, 11.85), and social media are convenient vs. not convenient (AOR = 6.17, 95% CI: 3.49, 10.92). (Table 4).

### 3.5. Predictors of HPV Self-Testing

An assessment of participants’ knowledge about self-testing revealed only 30.31% were aware of self-screening. When participants were asked to indicate their preference for self-screening, 42.91% reported they preferred self-testing, 47.24% preferred physician-performed screening, and 9.84% reported no preference. Almost two-thirds (57.48%) reported that they had an intention to take HPV self-sampling if they get self-sampling kits (see Table 1).

After controlling for the covariates, women’s intention to participate in self-screening was associated with their perception that social media provide privacy vs. no privacy (AOR = 2.67, 95% CI: 1.59, 4.48), social media provide social support vs. no social support, (AOR = 2.58, 95% CI: 1.53, 4.33), social media are less costly vs. costly (AOR = 3.67, 95% CI: 2.15, 5.26), and social media are convenient vs. less convenient (AOR = 4.17, 95% CI: 2.42, 7.17) (see Table 4).

## 4. Discussion

We examined the feasibility of MUW’s participating in future social media HPV self-testing-related studies. We also identified barriers and facilitators of social media utilization. Several findings from our study are worth noting. First, we found that over 32% of the women had never had a pap test or were unsure if they had had a pap test. Moreover, over 44% of the participants were overdue for their pap smears. These findings are consistent with previous studies reporting low cervical cancer screening rates among low-income women in the U.S. [39,40]. A little more than half of the MUW in our study indicated they would participate in a future social media and HPV self-testing study. Women between the ages of 30 and 49 years old were more likely to participate in HPV self-testing.

These findings imply that the cervical cancer screening rate among MUW continues to be low and remains a public health concern. Consistent and concerted efforts are needed to reach MUW with HPV self-testing information to increase cervical cancer screening rates. Targeted public health programs need to be directed to these hard-to-reach populations.

### 4.1. Self-Screening

Second, while most of the women (69.68%) were unaware of the HPV self-sampling screening, 42.9% of them preferred HPV self-screening compared to 47.2% who preferred physician-performed screening. These findings are different from other studies that reported high knowledge about self-screening and a high preference for self-screening [41,42]. In fact, a recent meta-analysis of 37 studies among 18,516 women from 24 countries across 5 continents indicated not only strong acceptance of self-sampling but also a strong preference for self-sampling over clinician sampling [43]. It is possible that because women in our current study were not aware of HPV self-screening (i.e., almost 70%), this may explain the reason why they reported a low preference for HPV self-screening. Financial issues play an important role in whether women are screened for cervical cancer. However, in the U.S., low-income women are eligible for Medicaid, which covers cervical cancer screening expenses [44], so lack of screening may be due in part to a lack of knowledge about screening. Interventions should be designed to create awareness about self-screening and also emphasize the importance and efficacy of self-screening. Making HPV screening information and HPV self-testing kits available to MUW may create awareness about HPV self-screening and subsequently increase cervical cancer screening behavior.

### 4.2. Social Media Usage

Third, we found a high percentage and frequent usage of social media among the study population. Widows and unemployed women reported regular use of social media more than married and employed women. Despite the low socioeconomic status (8 in 10 unemployed with yearly income below $20,000), we found that the majority of participants (82%) reported regular usage of social media, including Facebook, Instagram, Twitter, and WhatsApp (text messaging). While past literature indicates people with low incomes are less likely to have access to the internet and mobile phones, [45,46] our findings differ from those conclusions. It is possible that because social media usage has become popular [47], more people, irrespective of their socioeconomic conditions, are using mobile technology for social interactions, which is good for mHealth interventions. In the U.S., mobile technologies have bridged the digital divide [48] and low-income African-Americans and Hispanics are just as likely as Whites to own a mobile phone and use it for a wide range of activities [48]. In Quintiliani et al.’s [49] survey among females living in public housing, the researchers found that nearly all participants reported mobile phone usage for calls (97%) and text messages (84%). They found that most of the women living in public housing use internet (65%), social media (59%), and email (28%); and 70% had a Facebook account and 12% a Twitter account [49]. These data support our findings that social media usage is increasingly becoming popular even among MUW, including low-income populations and racial and ethnic minorities. Additionally, our findings show that 1 in 2 of the participants reported the likelihood of participating in the HPV self-testing interventions if such an educational program is offered on social media platforms (i.e., Facebook, Instagram, WhatsApp, and Twitter). These findings suggest that the use of social media to promote screening among low-income women is feasible, so it should be encouraged. While social media (i.e., mHealth application) could not have been possible one or two decades ago, now implementing mHealth intervention among MUW is becoming more feasible [27,28,29]. Using social media to promote HPV self-testing will be cost-effective and convenient because there is compelling evidence that mobile technology is a promising communication tool that can be used to reach the unreached MUW.

### 4.3. Barriers and Facilitators

Fourth, we identified factors that contribute to and/or hinder social media usage among MUW. The factors that influence women’s utilization of social media include privacy, social support, cost, and convenience of use, which are consistent with previous studies [50,51,52]. The barriers to social media usage reported by the participants included misinformation delivered through social media, social media can be time-consuming and distracting, social media provide insufficient information, and confidentiality concerns about social media usage. Our current findings support several other studies reporting similar conclusions. Lack of time and effort, lack of motivation and discipline, and confidentiality concerns are some of the reported barriers to mHealth utilization [50,51,52]. In our study, women who believed social media provide privacy (vs. no privacy) and social support (vs. no social support) were 6.23 and 7.18 times more likely to report a willingness to participate in a social media self-screening-related intervention, respectively. Studies have found that social support is a catalyst for behavior change broadly [53,54,55], and social media can provide informational and emotional support [56]. Further, we found that women who believe social media are less costly than other sources of information and that social media are convenient were 6.71 times and 6.17 times more likely to participate in a social media HPV screening study. Similarly, women who believed social media provide privacy, social support, and are convenient and less costly reported an intention to take HPV self-screening. The implication of our findings about facilitators and barriers to mHealth utilization is that intervention could be implemented to emphasize factors that facilitate social media utilization and also address the concerns about barriers to social media usage. With the proliferation of misinformation on social media and the internet, it is increasingly becoming difficult for people to trust social media as a credible outlet for health information [57]. Therefore, public health professionals may have to do more than just providing evidence and facts about screening to MUW. Trust-building such as more engagement (including listening, patience, and respect) with MUW may help address the misinformation and distrust in social media.

### 4.4. Limitations and Strengths

While the results of our study were encouraging and informative for future work, the study was not without limitations. The data were collected using self-report surveys, which could introduce social desirability, and the responses may not represent the participants’ lived experiences. Again, the participants were asked to recollect their social media usage and HPV self-testing behaviors and it is possible that the participants might have difficulty recollecting those pieces of information. The sampling selection could also introduce selection bias because the study did not incorporate random selection and thus the results cannot be generalized to other populations. The use of the convenient sample is another limitation, as the sample may not be representative of the target population. However, we had a fair representation of women from diverse racial and ethnic backgrounds in the study, with Caucasian, Hispanic, Black/African American, and others (Asian, American Indian) representing 40.16%, 28.74%, 27.17%, and 3.94%, respectively. Our study demographic distribution reflects the demographic distribution of McLennan County, where the study took place [58].

Despite these limitations, the study has several strengths. First, the data support the feasibility of promoting cervical cancer self-screening among underrepresented populations using social media platforms. Second, there is paucity in the existing literature to help us understand MUW’s social media usage behavior, their preference for HPV self-testing, and willingness to participate in future social media and HPV self-study. However, this study seeks to close the gap in the literature. Finally, the women in our study are among the hard-to-reach population for any kind of study. To be able to recruit an impressive number (*n* = 254) of the MUW to help us determine the feasibility of implementing social media intervention demonstrates the strength of the study.

## 5. Conclusions

In conclusion, the findings from the current study underscore the low cervical cancer screening rate among MUW and the need for researchers and healthcare professionals to increase their efforts to reach MUW for screening. While barriers to healthcare and accurate information exist among this population, frequent social media usage among LIW and their willingness to participate in social media HPV self-testing-related intervention studies make an HPV screening social media-driven campaign a viable option. Future social media interventions to promote HPV self-testing among low-income populations should highlight that social media provide privacy, provide social support, are less costly, and are convenient, the major reasons why 8 in 10 of the women in this study were using social media. Ultimately, research with larger samples should be conducted to further unpack the findings in this study.

## Figures and Tables

**Table 1 ijerph-18-10820-t001:** Demographic characteristics, screening behavior, knowledge, and preference among the participants (*n* = 254).

	Frequency	Percent
Age		
30–49	123	48.43
50–65	131	51.57
Marital Status		
Married	99	38.98
Living as married	14	5.51
Divorced	48	18.90
Widowed	34	13.39
Separated	24	9.45
Single, never been married	35	13.78
Race/Ethnicity		
White	102	40.16
Black	69	27.17
Hispanic	73	28.74
Other (Native American, Asian)	10	3.94
Income Level		
$0–$9999	140	55.12
$10,000–$19,999	72	28.35
$20,000 and above	42	16.54
Employment		
No	204	80.31
Yes	50	19.69
Pap-test/HPV Test		
Yes	172	67.72
No	31	12.20
Not sure	51	20.08
Status of Pap Test		
Current	141	55.51
Overdue	113	44.49
Intention for HPV Test		
Yes	146	57.48
No	108	42.52
Knowledge about HPV Self-Testing		
Yes	77	30.31
No	117	69.69
HPV Test Preference		
No preference	25	9.84
Self-Testing	109	42.91
Physician-performed Test	120	47.24

**Table 2 ijerph-18-10820-t002:** Multinomial logistic regression models of social media usage, intention to participate in social media study, and intention to participate in HPV self-testing by selected covariates (*n* = 254).

	SMU OR (95% CI)	Intention to Participate in SM Study OR (95% CI)	Intention to Participate in HPVST OR (95% CI)
Age			
30–49	1.89 (0.97–3.69)	1.81 (1.09–3.01) *	1.02 (0.61–1.70)
50–65	Ref (--)	Ref (--)	Ref (--)
Marital Status			
Single/Never married	1.63 (0.56–4.73)	0.46 (0.20–1.06)	0.92 (0.41–2.09)
Living as married	2.11 (0.49–9.10)	1.28 (0.39–4.25)	2.22 (0.63–7.81)
Divorced	2.53 (0.97–6.58)	1.09 (0.52–2.30)	2.39 (1.08–5.31) *
Widowed	2.96 (1.11–7.78) *	0.60 (0.27–1.33)	0.44 (0.19–1.01)
Separated	1.63 (0.49–5.43)	1.26 (0.49–3.28)	1.95 (0.72–5.30)
Married	Ref (--)	Ref (--)	Ref (--)
Race/Ethnicity			
Other (Native American, Asian)	0.48 (0.06–4.22)	0.93 (0.24–3.58)	2.48 (0.55–11.09)
Black	1.62 (0.72–3.64)	1.10 (0.58–2.10)	1.34 (0.69–2.62)
Hispanic	1.53 (0.66–3.57)	0.77 (0.40–1.48)	1.13 (0.58–2.20)
White	Ref (--)	Ref (--)	Ref (--)
Income Level			
$0–$9999	0.51 (0.16–1.62)	1.59 (0.76–3.33)	1.15 (0.54–2.42)
$10,000–$19,999	1.09 (0.52–2.25)	0.97 (0.44–2.15)	1.00 (0.44–2.24)
$20,000 and above	Ref (--)	Ref (--)	Ref (--)
Employment			
No	4.51 (1.33–5.23)	0.86 (0.46–1.62)	1.26 (0.67–2.38)
Yes	Ref (--)	Ref (--)	Ref (--)
Insurance			
No	0.75 (0.38–1.49)	1.05 (0.62 –1.76)	1.05 (0.62–1.76)
Yes	Ref (--)	Ref (--)	Ref (--)

Note: Ref = reference; OR = Odds ratio; SMU = Social media usage; CI = Confidence interval; SM = Social media; HPVST = HPV self-testing. * *p* < 0.05.

**Table 3 ijerph-18-10820-t003:** Distribution of participants’ past social media usage behavior, likelihood to use social media, facilitators, and barriers to social media usage (*n* = 254).

	Frequency (%)	
Past Social Media Usage		
None	46 (18.11)	
Facebook only	141 (55.51)	
Two Social Media *	38 (14.96)	
Three Social Media **	17 (6.69)	
WhatsApp (text messaging)	10 (3.94)	
Other (Twitter, Instagram)	2 (0.78)	
Social Media most likely to be used		
None	55 (21.65)	
Facebook only	157 (61.81)	
Two or more social media	42 (16.14)	
Likelihood of participating in social media study		
Yes	133 (52.36)	
No	71 (27.95).	
Not sure	50 (19.69)	
Comfortable participating in social media study		
Yes	140 (55.12)	
No	114 (44.88)	
Participate in Social Media		
Yes	123 (48.43)	
No	131 (51.57)	
	Yes *n* (%)	No *n* (%)
Facilitators of Social Media Usage		
social media provides privacy	129 (50.79)	125 (49.21)
social media provides social support	139 (54.72)	115 (45.28)
social media is less costly	143 (56.30)	111 (43.70)
social media is convenient	148 (58.27)	106 (41.73)
Barriers to Social Media Usage		
misinformation on social media	84 (33.07)	170 (66.93)
social media is time consuming or distracting	79 (31.10)	175 (68.90)
social media provides insufficient information	89 (35.04)	165 (64.96)
confidentiality concerns about social media	135 (53.15)	119 (46.85)

* Facebook/Instagram or Facebook/Twitter or Facebook/WhatsApp; ** Facebook/Instagram/Twitter or Facebook/Instagram/WhatsApp or Facebook/WhatsApp/Twitter.

**Table 4 ijerph-18-10820-t004:** Multinomial Logistic Regression Model of Likelihood of participating in social media self-testing and intention for self-testing by selected demographic characteristics (*n* = 254).

	**B**	**Std. Error**	**Wald**	**AdjOR (95% CI)**	** *p* ** **-Value**
Predictors of social media study participation					
Age	0.30	0.25	1.44	1.34 (0.83–2.17)	0.23
Race	0.14	0.16	0.85	1.16 (0.85–1.57)	0.36
Employment	0.06	0.36	0.02	1.06 (0.52–2.57)	0.88
Insurance	0.03	0.30	0.01	1.03 (0.57–1.87)	0.92
Income	0.44	0.21	4.39	1.55 (1.03–2.33)	0.04
Marital Status	0.13	0.08	2.56	1.14 (0.97–1.33)	0.11
Confidentiality (vs no confidentiality)	1.83	0.29	40.91	6.23(3.56–10.92)	0.00
Social support (vs. no social support)	1.97	0.30	44.61	7.18(4.03–12.80)	0.00
Less costly (vs costly)	1.90	0.29	42.99	6.71(3.80–11.85)	0.00
Convenience (vs. less convenience)	1.82	0.29	39.12	6.17(3.49–10.92)	0.00
Misinformation (vs. less misinformation)	0.53	0.27	3.70	1.70 (0.99–2.91)	0.05
Time-consuming (vs. less time consuming)	0.13	0.28	0.23	1.14 (0.66–0.63)	0.63
Inefficient (vs. efficient)	0.20	0.27	0.54	1.22 (0.72–2.07)	0.46
Privacy concerns (vs no priv. concerns)	−0.15	0.29	0.29	0.86 (0.49–1.50)	0.59
Predictors for Intention to conduct HPV self-screening					
Age	−0.16	0.24	0.44	085 (0.54–1.36)	0.85
Race	0.03	0.15	0.03	1.03 (0.76–1.38)	1.03
Employment	−0.34	0.36	0.91	0.71 (0.35–1.43)	0.71
Insurance	−0.17	0.29	0.34	0.84 (0.47–1.50)	0.84
Income	0.34	0.20	2.91	1.40 (0.95–2.07)	1.40
Marital Status	0.01	0.08	0.01	1.01 (0.87–1.17)	1.01
Privacy (vs no privacy)	0.98	0.26	13.76	2.67 (1.59–4.48)	0.00
Social support (vs. no social support)	0.95	0.27	12.80	2.58 (1.53–4.33)	0.00
Less costly (vs costly)	1.30	0.27	22.82	3.67 (2.15–6.26)	0.00
Convenience (vs. less convenience)	1.43	0.28	26.48	4.17 (2.42–7.17)	0.00
Misinformation (vs. less misinformation)	−0.31	0.27	1.33	0.73 (0.43–1.24)	0.25
Time consuming (less time consuming	−0.33	0.28	1.42	0.72 (0.42–4.24)	0.23
Inefficient (vs. efficient)	−0.43	0.27	2.53	0.65 (0.39–1.10)	0.11
Confidentiality (vs no Confidentiality)	0.45	0.26	3.07	1.58 (0.95–2.62)	0.08

Note: B = unstandardized coefficient; std error = standard error of the coefficient; AdjOR = Adjusted Odds Ratio. CI = Confidence Interval. Significant value *p* < 0.001. Adjusted covariates include age, marital status, race, employment, insurance, and income. coefficient.

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
