# Peer review of "Feasibility of Utilizing Social Media to Promote HPV Self-Collected Sampling among Medically Underserved Women in a Rural Southern City in the United States (U.S.)"

_ijerph, 2021, doi:10.3390/ijerph182010820_

Round 1

Reviewer 1 Report

Feasibility of Utilizing Social Media to Promote HPV Self-collected Sampling among Medically Underserved Women in a Rural Southern City

The paper is interesting and bring some ideas and propose some analysis on how to create awareness and participation to cervical self-screening in low income women (LIW) using social media which should be a good way to reach underserved-women.

Nevertheless, the study should be stronger through a randomised controlled trial where participation could be measured between women receiving social media messages and women who do not receive any message or receiving messages through another medium.

Furthermore, I would suggest not to evaluate WhatsApp with Facebook/Instagram together as it is not comparable to directly receive a message on the phone or to see an advertisement/message on Facebook or Instagram. That two different media should be evaluated separately to point out their strengths and weaknesses in creating awareness on cervical cancer self-screening.

To strengthen the study, the participation to self-screening should be evaluated instead of the intention to participate.

To sum-up, this paper contains some preliminary analysis on the acceptability for women to receive cervical cancer self-screening through social media but those analysis should be reinforced using stronger designs than self-reported questionnairse and intention to participate.

Reviewer 2 Report

Dear Authors,

Thank you very much for this opportunity to review the manuscript. Overall the content is great, but I have concerns how did the authors explain the usefulness of HPV-self sampling against the participants? The test might influence only women's anxiety. Were all participants able to interpret the meaning of the test? 

Reviewer 3 Report

Dear Authors, I am commenting here on your brilliant and curious work with the only purpose of trying to improve it with my suggestions:

 Title: Please, mention the country in the title.

Line 21 of Abstract. I think is HPVST instead of HVST.

Last lines of the introduction (from line 83 to 89): Personally, I prefer a classic wording where the objective is defined and then the authors explain what is done to achieve the objective. The way of writing leaves it up to the reader, what the possible objective would be.

“Understanding the frequency of MUW’s social media usage and HPV self-testing behaviors and identifying factors influencing their social media usage will help in developing a tailored intervention to promote cervical cancer screening among MUW” is a hypothesis; it should be mentioned before the objective.

Regarding the sample size. Please, recognize the work of our statistical colleagues. Please reference the G*Power Software: “Faul, F., Erdfelder, E., Lang, A.-G., & Buchner, A. (2007). G*Power 3: A flexible statistical power analysis program for the social, behavioral, and biomedical sciences. Behavior Research Methods, 39, 175-191”

If the MDPI editors have not already told you, the registration number and the date of approval of the study by the university’s Institutional Review Board should be included in the document.

From line 114 to line 120, please check the quotation marks and punctuation. “The items were: (a) “I intend… self-sampling kits”, measured … (b) “if HPV self-sampling… participate in that program”, measured…

Check table 2, a parenthesis is missing.

It is not clear to me where the results of the dependent variables are reported: “I intend to take the HPV self-sampling test if I get self-sampling kits” and “if HPV self-sampling test education is provided on any social media platform (Facebook, Instagram, WhatsApp, Twitter, text messaging), I will be … to participate in that program”.  I assume it is not “HPV test preference” from table 1 or " Likelihood or Comfortable participating in social media study " from table 2.   It should be remembered that a preference for one type of test or another does not mean a predisposition to take a test.

In summary, the study is brilliant, but there is a lot of confusion in the text with the two dependent variables, it is not clear to which variable the multinomial logistic regression model has been applied. I would ask the authors to make one last effort to clarify their work so that it can be more easily cited. 

Just one more question, if there are 39% of women without health insurance, what happens in the US if a woman has a positive Pap smear or HPV test result, is she followed up, biopsied or treated? Because if women do not have access to these services, what is the point of this study? Just curious, what happens to low-income women living in the US if they test are positive? “If I don't have the money to pay for HPV treatment, why do I want to know if I have HPV?”, individualistic thinking is widely held in US society.

Congratulations for your study

Reviewer 4 Report

1. Please mention the statistical test used for sample size determination using G*Power software. 2. The authors are not mentioned anywhere in the manuscript on how big their model is. 3. They haven’t defined how individual variates influence the model. 4. The authors mentioned that they adjusted the covariates' age, marital status, race, employment, insurance, and income. How these covariates are influencing the model. 5. How the screening rate among MUW is influenced by these covariates is not described anywhere in the manuscript. 6. The manuscript is confusing and needs to be rewritten clearly.

Reviewer 5 Report

The authors present a well written manuscript on a study, which has evaluated the feasibility of medical underserved women’s (MUV) participating in future social media HPV self-testing-related studies including barriers and facilitators of social media utilization. It shows that the cervical cancer screening rate among MUW is low and remains a public health concern. Consistent and concerted efforts are needed to reach MUW with HPV self-testing information to increase cervical cancer screening rates.

This information is timely and generally relevant to reduce cervical cancer.

My only proposal is to better justify the statement on high sensitivity of self-test HPV in the diagnosis of cervical cancer (lines 63-64).
